Computed tomography-based radiomic features combined with clinical parameters for predicting post-infectious bronchiolitis obliterans in children with adenovirus pneumonia: a retrospective study

Zhang Li 1
He Ling 1
Zhang Guangli 2
Tian Xiaoyin 2
Wang Haoru 1
Wang Fang 3
Chen Xin 1
Zheng Yinglan 1
Li Man 3
Li Yang 3
Luo Zhengxiu 2 luozhengxiu816@hospital.cqmu.edu.cn
1 Department of Radiology Children’s Hospital of Chongqing Medical University, National Clinical Research Center for Child Health and Disorders, Ministry of Education Key Laboratory of Child Development and Disorders, Chongqing Key Laboratory of Pediatric Metabolism and Inflammatory Diseases , Chongqing , China
2 Department of Respiratory Medicine, Children’s Hospital of Chongqing Medical University, National Clinical Research Center for Child Health and Disorders, Ministry of Education Key Laboratory of Child Development and Disorders, Chongqing Key Laboratory of Pediatric Metabolism and Inflammatory Diseases , Chongqing , China
3 Department of Research and Development, Shanghai United Imaging Intelligence Co., Ltd. , Shanghai , China
Guan Fanglin
Electronic publication date: 2025 Mar 31
Publication date: 2025
Volume: 13
Electronic Location ID: e19145
Received 2024 Nov 17; Accepted 2025 Feb 19
Copyright: © 2025 Zhang et al.
Copyright year: 2025
Copyright holder: Zhang et al.
License: This is an open access article distributed under the terms of the Creative Commons Attribution License, which permits unrestricted use, distribution, reproduction and adaptation in any medium and for any purpose provided that it is properly attributed. For attribution, the original author(s), title, publication source (PeerJ) and either DOI or URL of the article must be cited.
License URL: https://creativecommons.org/licenses/by/4.0/

Keywords: Children, Bronchiolitis obliterans, Adenovirus pneumonia, Radiomics, Prediction

Funding: Project of Young and Middle Medical Distinguished Team in Chongqing, China This work is supported by The Project of Young and Middle Medical Distinguished Team in Chongqing, China. The funders had no role in study design, data collection and analysis, decision to publish, or preparation of the manuscript.

==============================
Objectives

To develop a model incorporating computed tomography (CT) radiomic features and clinical parameters for predicting bronchiolitis obliterans (BO) with adenovirus pneumonia in children.

Methods

A total of 165 children with adenovirus pneumonia between October 2013 and February 2020 were enrolled retrospectively. Among them, BO occurred in 70 patients, and the remaining 95 patients did not have BO. These children were stratified into training and testing groups at a ratio of 7:3. Manual segmentation of lesions in baseline CT images during acute pneumonia was performed to extract radiomic features. Multiple statistical methods were used to determine the best radiomic features. Combined models based on radiomic and clinical features were established via logistic regression (LR), random forest (RF), and support vector machine (SVM) algorithms. Model performance was evaluated via the area under the receiver operating characteristic curve (AUC).

Results

A total of 2,264 radiomic features were extracted from the lesions, from which 10 optimal radiomic features were ultimately selected. The length of hospitalization, number of pneumonia lobes, and optimal radiomic features were incorporated into the combined models. In the training group, the AUCs of the combined LR, RF and SVM models were 0.946, 0.977, and 0.971, respectively; while in the testing group, they yielded AUCs of 0.890, 0.859, and 0.885, respectively. The predictive performance of these combined models surpassed that of the radiomic and clinical models.

Conclusion

Combining CT-based radiomic features with clinical parameters can offer an effective noninvasive model to predict BO in children with adenovirus pneumonia.

Introduction

Bronchiolitis obliterans (BO) is a clinical syndrome characterized by persistent airflow obstruction resulting from chronic inflammatory damage to small airways. Severe lower respiratory tract infection, connective tissue disease, organ or bone marrow transplantation, toxic inhalation, gastroesophageal reflux, and certain drugs can induce BO (Jerkic et al., 2020; Hakim et al., 2019). Post-infection bronchiolitis obliterans (PIBO), which arises from a consequence of respiratory tract infection, represents the most prevalent cause of BO among pediatric patients (Onay et al., 2020). Children’s respiratory systems are not fully developed and their bronchioles are more sensitive and vulnerable. Therefore, when the respiratory system of children is infected or injured, it is more likely to cause inflammation and damage of the bronchioles and thus develop PIBO. Adenovirus is recognized as the primary pathogen associated with PIBO in children, with a study reporting that 65.9% of children with acute lower respiratory tract adenovirus infections developed BO (Huang et al., 2021). Unfortunately, many children who develop BO subsequent to adenovirus pneumonia experience frequent readmissions due to recurrent respiratory infections and exhibit diminished quality of life (Colom et al., 2015). Hence, early diagnosis of BO after adenovirus pneumonia is crucial for clinical treatment and improving the prognosis of affected children.

Several clinical models have been developed to predict BO after adenovirus pneumonia (Colom & Teper, 2009; Zhong, Lin & Dai, 2020). However, these models exhibit low sensitivity. Imaging examination is one of the main bases for judging the severity and evaluating the prognosis of adenovirus pneumonia. However, traditional imaging diagnosis methods often rely on the experience of doctors and manual interpretation of medical images, which is not only inefficient, but also may have subjectivity and error. We hope to establish a more accurate and objective method to provide more diagnostic information for early prediction of adenovirus PIBO in children. Radiomics, an emerging field of research focused on extracting radiomic features from medical images (Mayerhoefer et al., 2020), can provide a promising approach. These radiomic features acquire tissue and lesion attributes such as heterogeneity and shape, which can be utilized independently or in conjunction with demographic, histologic, genomic, or proteomic data to address clinical challenges (Mayerhoefer et al., 2020; Chen et al., 2021b). Radiomics has played a pivotal role in the research of lung diseases. In adults, radiomics has been used to identify different types of interstitial pneumonia (Park et al., 2011), differentiate between pneumonia and lung tumors (Zhang et al., 2019), and predict the prognosis of lung tumors (Huang et al., 2016; Chen et al., 2019; van Timmeren et al., 2017). In children, CT features have also been utilized to differentiate between different types of pneumonia (Wang et al., 2019) and facilitate early identification of necrotizing pneumonia (Chen et al., 2021a). Studies have shown that radiomics plays an essential role in differentiating COVID-19 from other forms of pneumonia (Peng et al., 2022; Cardobi et al., 2021; Al-Kuraishy et al., 2022), as well as in predicting the severity, progression and outcome of patients with COVID-19 (Shi et al., 2022; Wang et al., 2021). Radiomics has been widely studied in lung diseases, but few studies have explored the potential use of radiomics in forecasting BO after adenovirus pneumonia. Therefore, this study aimed to extract radiomic features from the CT images of children with adenovirus pneumonia and integrate them with clinical parameters to establish a machine learning model for predicting the occurrence of PIBO, in order to predict the occurrence of PIBO early, help clinical treatment of children with adenovirus pneumonia early, improve the prognosis of children with adenovirus pneumonia, and improve the quality of life of children with adenovirus pneumonia.

Methods

Patients

The research received approval from the Ethics Committee of Children’s Hospital of Chongqing Medical University (No.2023–151). Informed consent was waived due to the retrospective nature of the study. Data collection for this study began on April 3, 2023.

We enrolled children with adenovirus pneumonia who were hospitalized between October 2013 and February 2020. Inclusion criteria were: (i) aged 1 month to 18 years, (ii) acute lower respiratory symptoms, lung infltration and tested positive for adenovirus DNA in nasopharyngeal secretions or by direct immunofluorescence antigen test (Zhong, Lin & Dai, 2020), and (iii) had CT images acquired during acute adenovirus pneumonia prior to treatment. Exclusion criteria were: (i) incomplete clinical information, (ii) absence of CT images during acute adenovirus pneumonia or poor-quality CT images, (iii) concurrent infection with other pathogens during adenovirus pneumonia before the diagnosis of BO, and (iv) a previous history of organ transplantation.

Two pediatric respiratory specialists (5 and 8 years of experience) independently diagnosed the enrolled patients either as BO or non-BO patients according to the diagnostic criteria. In cases of disagreement, a consensus was reached through in-depth discussions. The children were diagnosed with BO after adenovirus pneumonia on the basis of the following criteria (Colom & Teper, 2009; Aguerre et al., 2010): (i) a history of adenoviral pneumonia, (ii) persistent or recurrent wheezing, coughing, shortness of breath, dyspnea, and exercise intolerance. Extensive wheezing and moist rales were auscultated bilaterally for more than 6 weeks with little response to bronchodilators. (iii) High-resolution computed tomography (HRCT) revealed a mosaic pattern, bronchiectasis, and thickening of the bronchial wall. (iv) Pulmonary function tests indicated obstructive or mixed ventilatory dysfunction in small airways with predominantly negative results in bronchodilation tests. (v) Other chronic lung diseases that could induce wheezing and coughing, such as bronchial asthma, congenital bronchopulmonary dysplasia, tuberculosis, and cystic fibrosis, were ruled out.

The detailed patient selection process is illustrated in Fig. 1. Patient data were randomly divided into training group and testing group at a ratio of 7:3.

Figure 1 The flowchart of the inclusion of study subjects.

CT image acquisition

Chest CT data were acquired prior to treatment via LightSpeed VCT 64-slice spiral CT (GE Healthcare, Chicago, IL, USA), Brilliance iCT 256-slice spiral CT (Philips Medical System, Eindhoven, Netherlands), Optima CT660 128-slice spiral CT (GE Healthcare, Chicago, IL, USA) scanners. The scanning parameters included tube voltage ranging from 80 to 100 kV, automatically regulated tube current, pitch of 0.984:1, collimation of 0.6 mm, slice thickness and slice interval of 5.0 mm, reconstructed slice thickness of either 1.25 or 1.0 mm, and a reconstructed slice interval of either 1.25 or 1.0 mm. The data were reconstructed via a conventional algorithm.

Clinical data

Clinical data were extracted from medical records by trained research staff via a standard collection form. The collected data included demographic characteristics, duration of fever (in days), length of hospitalization (in days), peak temperature (in °C), laboratory findings and clinical treatment.

Two experienced radiologists, with 6 and 12 years of experience, independently evaluated imaging features including the number of pneumonia lobes, pulmonary consolidation lobes, and the presence or absence of pleural effusion. In case of disagreement, consensus was reached through discussion.

Image segmentation

Digital Imaging and Communications in Medicine (DICOM) images were exported from the Picture Archiving and Communication System (PACS) and regions of interest for pulmonary lesions were manually delineated using ITK-SNAP (Version 3.8) software. The lesion margins on thin-slice CT images were manually delineated by a radiologist with 6 years of experience on the axial lung window (Fig. 2), and verified by a senior radiologist with 12 years of experience.

Figure 2 The example of manually segmenting and contouring ROIs.

(A and B) A 21-month-old male in the PIBO group; (C and D) an 81-month-old male in the non-PIBO group.

The lesion margins were manually delineated of thin-slice CT images by a radiologist with 6 years of experience. The segmentation results were then verified by a senior radiologist with 12 years of experience.

To ensure the availability of radiomic features due to the subjective nature of manual delineation (Fornacon-Wood et al., 2020), ROIs were redelineated for 30 randomly selected patients (BO: non-BO = 12:18). The intraclass correlation coefficient (ICC) was calculated for all extracted radiomic features from the two delineated RIOs in these 30 patients, and the features with an ICC > 0.85 were included in the subsequent study.

Radiomics feature extraction and selection

We uploaded the images and corresponding ROIs to the uAI research portal (uRP (Wu et al., 2023), Version: 20220915, United Imaging Intelligence, China), applied a series of preprocessing steps to ensure consistency and quality in radiomic feature extraction: we used the formula f(x)=s(x−μx)δx, x and f(x) are the original intensity and normalized intensity, μx and δx are the mean and standard deviation of the image intensity values, and s is the optimal scaling defined by scale. Images were normalized centered on the standard deviation of the mean, resampled to voxel size 1 * 1 * 1 mm3 using b-spline interpolation, intensity normalization with a fixed bin width 25 Hounsfield units (HUs), and Z score normalization to obtain a standardized normal distribution of image intensities. Radiomic features such as first-order features, shape features, and texture features were extracted via PyRadiomics embedded within the uRP. The texture features included the gray-level co-occurrence matrix (GLCM), gray-level run-length matrix (GLRLM), gray-level dependence matrix (GLDM), neighborhood gray-tone difference matrix (NGTDM), and gray-level size-zone matrix (GLSZM).

To improve model performance and reduce overfitting in subsequent data analysis processes, it is essential to select the best feature set. We employed multivariate information (MI) and minimum redundancy maximum relevance (mRMR) methods to reduce redundant features. Subsequently, we selected non-zero coefficient features using the least absolute shrinkage and selection operator (LASSO), which employs an L1 regularizer as the cost function along with 10-fold cross validations. Ultimately, we selected the most relevant predictor features associated with prognosis. The radiomics score (Radscore) for each patient used in model construction was computed by linearly multiplying the selected features (F) with their corresponding coefficients (β) (Radscore = β0 + β1 * F1 + β2 * F2 + … + βn * Fn).

Models construction and evaluation

We constructed radiomic models using the Radscore, clinical models based on selected clinical parameters via multivariate logistic regression analysis in the training group, and combined models by integrating radiomic and clinical models. Logistic regression (LR), support vector machine (SVM), and random forest (RF) methods were utilized for model construction.

Model performance was assessed via the area under the receiver operating characteristic curve (AUC), and corresponding metrics, such as sensitivity, accuracy, specificity, positive predictive value (PPV), negative predictive value (NPV) and F1 score, were calculated. Calibration curves were plotted to assess model calibration. Decision curve analysis (DCA) was used to evaluate net benefit at different threshold probabilities and assess clinical utility. Model comparison was conducted via the Delong test, net reclassification improvement (NRI), and integrated discrimination improvement (IDI). Finally, a nomogram based on the best-performing model was created.

Statistical analysis

Statistical analysis was conducted via R statistical software (version: 4.1.1). All the statistical tests were two-sided, and a significance level of p < 0.05 was considered as statistically significant. Quantitative characteristics were evaluated via the Mann‒Whitney U test and are presented as medians with their corresponding quartiles. Qualitative characteristics were examined via the chi-square test and reported as counts (%). Influence characteristics that exhibited statistical significance at p < 0.05 in the univariate logistic analysis were included in the multivariate analysis using a stepwise selection approach.

Results

Patient characteristics and clinical model construction

During the study period, data on 2,116 patients diagnosed with adenovirus pneumonia were collected. Ultimately, 165 patients who satisfied the inclusion and exclusion criteria were included in the study. Among them, 70 patients belonged to the BO group while 95 patients were part of the non-BO group. The participants were divided into a training group (n = 115, BO: non-BO = 49:66) and a testing group (n = 50, BO: non-BO = 21:29) at a ratio of 7:3. The detailed characteristics of the patients can be found in Table 1.

Table 1 Clinical parameters in training group and testing group.

Characteristics	Training cohort	Testing cohort	
Non-PIBO (n = 66)	PIBO (n = 49)	p value	Non-PIBO (n = 29)	PIBO (n = 21)	p value	
Gender (%)			0.006			0.991	
Female	28 (42.4)	9 (18.4)		11 (37.9)	8 (38.1)		
Male	38 (57.6)	40 (81.6)		18 (62.1)	13 (61.9)		
Age (Month)	20 [13.75–43.25]	13 [10–24]	0.005	21 [11–47]	13 [10.5–22.5]	0.053	
Duration of fever (Day)	10 [7–14]	16 12–21]	<0.001	9 [6.5–13]	15 [11–22.5]	0.002	
Peak temperature (°C)	40 [39.70–40.38]	40 [39.8–40.4]	0.552	40 [39.6–40.4]	40 [39.6–40.3]	0.913	
Length of hospitalization (Day)	8 [7–11]	15 [12–20]	<0.001	10 [7.5–13]	16 [11–30]	0.001	
Adenovirus load ( ×104)	3,335 [269.25–6,610]	7,840 [988–11,500]	0.001	6,260 [200–12,400]	6,560 [3,395–10,130]	0.716	
CRP (mg/L)	9.5 [8–19.25]	15 [8–23.5]	0.166	17 [8–32.5]	8 [8–23]	0.199	
PCT (ng/ml)	0.58 [0.19–1.64]	1.72 [0.36–3.2]	0.026	0.69 [0.14–1.68]	2.05 [0.54–2.95]	0.036	
WBC (×109/L)	7.43 [5.35–11.15]	7.36 [4.34–13.36]	0.919	7.56 [5.91–10.71]	7.65 [6.31–10.04]	0.930	
PLT (×109/L)	263 [215–370.75]	279 [202.5–460]	0.747	264 [185.5–353]	273 [199–362.5]	0.768	
Hb (g/L)	111.5 [104–119.25]	106 [90–114.5]	0.01	111 [104.5–120.5]	97 [88–109.5]	<0.001	
L (%)	0.40 [0.31–0.54]	0.34 [0.27–0.44]	0.071	0.34 [0.22–0.53]	0.32 [0.24–0.43]	0.687	
ALB (g/L)	38.5 [33.78–42.5]	35.3 [29.55–39.75]	0.003	38.9 [34.95–42.55]	33.5 [29.1–39.8]	0.009	
ALT (U/L)	21.95 [14.25–35.03]	28.9 [18.55–39]	0.062	24 [13.8–31.6]	36.3 [21.2–53.15]	0.016	
AST (U/L)	58.85 [39.08–83.48]	74.9 [42.9–105.4]	0.089	44.8 [32.7–69.6]	83.1 [57.6–131.6]	0.001	
LDH (U/L)	490.5 [331.75–725.4]	694.3 [400–1,134.45]	0.003	382 [299–600.5]	817 [514.3–1,497.35]	<0.001	
Number of pneumonia lobes	3 [2–4]	5 [4–5]	<0.001	3 [1.5–4]	5 [4–5]	0.002	
Number of pulmonary consolidation lobes	1 [1–2]	2 [1.5–3]	<0.001	1 [0–1]	3 [1.5–4]	<0.001	
Pleural effusion (%)			0.117			0.176	
Yes	19 (28.8)	21 (42.9)		6 (20.7)	8 (38.1)		
No	47 (71.2)	28 (37.3)		23 (79.3)	13 (61.9)		
Invasive mechanical ventilation (%)			<0.001			0.003	
Yes	4 (6.1)	19 (38.8)		3 (10.3)	10 (47.6)		
No	62 (93.9)	30 (61.2)		26 (89.7)	11 (52.4)		
CPAP (%)			<0.001			0.176	
Yes	8 (12.1)	26 (53.1)		6 (20.7)	8 (38.1)		
No	58 (87.9)	23 (46.9)		23 (79.3)	13 (61.9)		
Oxygen therapy (Day)	1 [1–1]	8 [45–12]	<0.001	5 [1–9]	11 [7–13.5]	<0.001	
Bronchoalveolar lavage (%)			0.021			0.077	
Yes	33 (50)	35 (71.4)		12 (41.4)	14 (66.7)		
No	33 (50)	14 (28.6)		17 (58.6)	7 (33.3)		
Use of IVIG (%)			<0.001			0.025	
Yes	14 (21.2)	27 (55.1)		10 (34.5)	14 (66.7)		
No	52 (78.8)	22 (44.9)		19 (65.5)	7 (33.3)		
Use of glucocorticoids (%)			0.001			0.416	
Yes	39 (59.1)	43 (87.8)		19 (65.5)	16 (76.2)		
No	27 (40.9)	6 (12.2)		10 (34.5)	5 (23.8)		
Notes:

CRP, C-reactive protein; PCT, procalcitonin; WBC, white blood count; PLT, Platelet; Hb, hemoglobin; L(%), lymphocyte percentage; ALB, albumin; ALT, glutamic-pyruvic transaminase; AST, glutamic-oxalacetic transaminase; LDH, lactate dehydrogenase; CPAP, Continuous positive airway pressure.

Several clinical characteristics, including age, gender, duration of fever, length of hospitalization, hemoglobin (Hb), albumin (ALB), lactate dehydrogenase (LDH), the number of pneumonia lobes, pulmonary consolidation lobes, the proportion of invasive mechanical ventilation (IMV), oxygen therapy, bronchoalveolar lavage, use of intravenous immunoglobulin (IVIG), and use of glucocorticoids were found to have statistically significant (p < 0.05) in the univariate analysis. Ultimately, length of hospitalization (OR = 1.203, p < 0.001) and number of pneumonia lobes (OR = 3.311, p < 0.001) were selected as clinical parameters via stepwise selection in multivariate logistic regression analysis (Table 2). These two variables were used to develop the clinical models. The AUCs of the LR, RF and SVM machine learning models were 0.878, 0.938, and 0.887, respectively, in the training cohort and 0.833, 0.849, and 0.804, respectively, in the testing cohort (Fig. 3).

Table 2 Univariate and multivariate logistic analysis in the cohort.

Characteristics	Univariate	Multivariate	
p value	OR (95% CI)	p value	OR (95% CI)	
Age (Month)	0.006	0.964 [0.940–0.990]			
Gender	0.008	0.305 [0.128–0.731]			
Duration of fever (Day)	<0.0001	1.190 [1.098–1.290]			
Length of hospitalization (Day)	<0.0001	1.293 [1.154–1.448]	0.0005	1.203 [1.084–1.334]	
Hb (g/L)	0.033	0.972 [0.948–0.998]			
ALB (g/L)	0.004	0.909 [0.852–0.971]			
LDH (U/L)	0.002	1.001 [1.000–1.002]			
Number of pneumonia lobes	<0.0001	3.598 [2.263–5.720]	<0.0001	3.311 [1.933–5.672]	
Number of pulmonary consolidation lobes	0.0001	3.598 [2.263–5.720]			
Invasive mechanical ventilation	0.0001	9.817 [3.068–31.409]			
Oxygen therapy (Day)	0.0002	1.184 [1.083–1.294]			
Bronchoalveolar lavage	<0.0001	8.196 [3.240–20.731]			
Use of IVIG	0.001	0.202 [0.075–0.540]			
Use of glucocorticoids	0.0003	4.558 [2.017–10.304]			
Notes:

OR, odds ratio; CI, confidence interval.

Figure 3 The ROC curves of the three classifiers in the training and testing groups.

In testing group, the combined model had the highest AUC value of 0.890 in LR classifier.

Radiomics features extraction, selection and model construction

A total of 2,264 radiomic features were extracted from the ROIs on the CT images of 115 patients in the training group. Among the 30 patients with redelineated ROIs, 2,179 features (96.25%) extracted from the two delineated ROIs had an ICC > 0.85 and were included in the subsequent analysis.

Through the MI, mRMR, and LASSO methods, 10 optimal features (Table S1) were ultimately selected (Fig. 4). These 10 features include two first-order features, seven texture features, and one shape feature. The texture features included three GLSZM, one GLDM, and three GLCM. The radiomic score (Radscore) was calculated for each patient, forming the basis for constructing radiomic models.

Figure 4 Radiomics features selection.

(A) Selection of parameter (α) using the LASSO (10-foldncross validation method), the best α value was 0.00955. (B) The LASSO coefficient profile was plotted using coefficients against log (α) = −2.02. (C) Violin plot of Radscores features between PIBO and non-PIBO group, Mann-Whitney U test was used for all Radiomics features. (D) The 10 optimal Radscores features were chosen and the corresponding coefficients were evaluated.

The AUCs of the LR, RF and SVM machine learning models were 0.891, 0.960, and 0.919, respectively, in the training group and 0.734, 0.765, and 0.739, respectively, in the testing group (Fig. 3).

Combined model construction and comparison

We constructed combined models integrating the radiomic and clinical models via the LR, RF and SVM machine learning models. The detailed evaluation indicators for these models in the training and testing groups are presented in Table 3. In the testing cohort, the LR machine learning model performed exceptionally well with AUC (95% CI) of 0.890 [0.792–0.988], sensitivity of 0.810, specificity of 0.897, accuracy of 0.860, PPV of 0.850, NPV of 0.867, and F1 score of 0.829, outperforming the other two models. The ROC curves for all three machine learning models in both the training group and testing group are shown in Fig. 3. The AUCs of the LR, RF and SVM machine learning models were 0.946, 0.977, and 0.971, respectively, in the training group and 0.890, 0.859, and 0.885, respectively, in the testing group. The LR machine learning model achieved an AUC of 0.946 in the training group and 0.890 in the testing group, outperforming those from both clinical and radiomics models.

Table 3 The detailed evaluation indicator information of three machine learning models in the training and testing groups.

Classifiers	Models	Group	AUC	Sensitivity	Specificity	Accuracy	PPV	NPV	F1-score	
LR	Clinical model	Training cohort	0.878 (0.815–0.941)	0.776	0.818	0.800	0.760	0.831	0.768	
Testing cohort	0.833 (0.707–0.959)	0.810	0.727	0.760	0.680	0.840	0.739	
Radscore model	Training cohort	0.891 (0.833–0.948)	0.857	0.818	0.722	0.793	0.698	0.824	
Testing cohort	0.734 (0.593–0.875)	0.762	0.724	0.640	0.588	0.667	0.664	
Combined model	Training cohort	0.946 (0.91–0.982)	0.816	0.879	0.852	0.833	0.866	0.825	
Testing cohort	0.890 (0.792–0.988)	0.810	0.897	0.860	0.850	0.867	0.829	
RF	Clinical model	Training cohort	0.938 (0.898–0.978)	0.857	0.864	0.861	0.824	0.891	0.840	
Testing cohort	0.849 (0.722–0.976)	0.810	0.690	0.740	0.654	0.833	0.723	
Radscore model	Training cohort	0.959 (0.929–0.990)	1.000	0.803	0.817	0.938	0.771	0.968	
Testing cohort	0.762 (0.627–0.897)	0.810	0.690	0.640	0.615	0.649	0.699	
Combined model	Training cohort	0.977 (0.956–0.998)	0.857	0.985	0.930	0.977	0.903	0.913	
Testing cohort	0.859 (0.755–0.963)	0.619	0.897	0.780	0.812	0.765	0.703	
SVM	Clinical model	Training cohort	0.887 (0.825–0.949)	0.633	0.924	0.800	0.861	0.772	0.729	
Testing cohort	0.804 (0.671–0.937)	0.667	0.862	0.700	0.778	0.781	0.718	
Radscore model	Training cohort	0.919 (0.869–0.968)	0.918	0.818	0.800	0.842	0.779	0.878	
Testing cohort	0.739 (0.599–0.879)	0.810	0.690	0.640	0.579	0.677	0.675	
Combined model	Training cohort	0.971 (0.943–0.998)	0.878	0.939	0.913	0.915	0.912	0.896	
Testing cohort	0.885 (0.787–0.983)	0.810	0.793	0.810	0.739	0.852	0.773	
Note:

AUC, Area under curve; LR, Logistic regression; RF, Random forest; SVM, Support vector machine; PPV, Positive predictive value; NPV, Negative predictive value.

The Delong test was utilized to compare the LR machine learning models’ performance (Table 4). The combined model demonstrated significantly better predictive performance than the Radscore model in both the training and testing cohorts (p = 0.031 in the training cohort and p = 0.023 in the testing cohort). In the training cohort, the combined model’s predictive ability was notably superior to that of the clinical model (p = 0.002), whereas no significant difference between the two models in the testing cohort (p = 0.154).

Table 4 The Delong test, net reclassification improvement (NRI) and integrated discrimination improvement (IDI) of LR model.

Compaired models	Group	Delong	p value	NRI	p value	IDI	p value	
Combined model vs.
Clinical model	Training cohort	3.061	0.002	1.349	<0.001	0.440	<0.001	
Testing cohort	1.425	0.154	1.343	<0.001	0.338	<0.001	
Combined model vs.
Radscore model	Training cohort	2.151	0.031	0.743	<0.001	0.175	<0.001	
Testing cohort	2.268	0.023	0.289	0.052	0.219	0.011	

We further assessed the LR machine learning model’s performance via net reclassification improvement (NRI) and integrated discrimination improvement (IDI) (Table 4). The IDI showed that the combined model outperformed both the clinical model and the radiomic model in both the training group and the testing group. In the testing group, the combined model had 0.338 (p < 0.001) better prediction accuracy than the clinical model and 0.219 (p = 0.011) better prediction accuracy than the radiomic model.

The Fig. 5 illustrates the calibration curve and decision curve analysis (DCA) of the LR machine learning model. The calibration curve indicates that the model’s predicted probabilities align well with actual probabilities, demonstrating good calibration. The DCA demonstrates that the combined model outperformed both the clinical and radiomic models in terms of predictive performance when considering a risk threshold between 0.2 and 1.0.

Figure 5 The calibration curves and decision curve analysis of the LR machine learning models in the training and testing group.

The X-axis of the calibration curve represents the predicted risk of developing PIBO, the Y-axis represents the actual diagnosed PIBO, the diagonal dashed line represents the perfect prediction of the ideal model, and the solid line represents the performance of the model, of which a closer fit to the diagonal dotted line represents a better prediction. The X-axis of DCA is the threshold probability, the Y-axis is the net benefit, the black horizontal lines and arcs are the reference lines, and the colored curve is the benefit curve. (A) The calibration curves in the training group; (B) The calibration curves in the testing group; (C) The DCA in the training group; (D) The DCA in the testing group.

Given that the LR machine learning model showed excellent prediction performance, we used two independent risk factors (the number of pneumonia lobes and the length of hospital stay) and the Radscore to construct a nomogram (Fig. 6) based on LR machine learning model to facilitate clinical decision-making.

Figure 6 Nomogram for predicting the probability of BO after adenovirus pneumonia.

The nomogram was drawn with two clinical parameters (number of pneumonia lobes and length of hospitalization) and radscore. The top portion of the diagram displays a points scale ranging from 0–100 for each variable. Scores for all three variables are combined and presented on a total points scale ranging from 0–180. The anticipated probability is determined by locating the corresponding position on the predicted value line perpendicular to the total points line.

Discussion

Adenovirus pneumonia is a common cause of PIBO in pediatric patients. This study examined the clinical information and imaging data of adenovirus pneumonia patients and established a combined model that integrates clinical and radiomic features to predict the occurrence of BO early after adenovirus pneumonia, the results of which demonstrated exceptional predictive efficacy. Our findings highlight the utility of CT-based radiomic analysis in providing additional prognostic information for BO prediction and enhancing the predictive performance of clinical models. These findings offer insights into the use of artificial intelligence for predicting the prognosis of children with adenovirus pneumonia.

Our findings revealed that the duration of hospitalization and the number of pneumonia lobes were identified as independent clinical risk factors for BO after adenovirus pneumonia, a conclusion that is consistent with prior studies (Murtagh et al., 2009; Castro Rodriguez et al., 2006). The duration of hospitalization and the severity of lung lesions can be regarded as proxies for pneumonia severity, implying a connection between the severity of adenovirus pneumonia and the likelihood of BO development. This aligns with the findings of Yalçin et al. (2003). Using these two variables, we developed clinical models that exhibited good predictive performance. Peng et al. (2023) previously developed a nomogram prediction model for PIBO in children with adenovirus pneumonia after invasive ventilation, achieving marvelous predictive performance with an AUC of 0.857, slightly outperforming our models. Such differences in model performance may stem from variations in the inclusion criteria or the specific variables analyzed. Peng et al. (2023) included children who developed PIBO after IMV under 3 years of age, and the variables used to develop the nomogram included gender, duration of fever, ADV load, and fungi coinfection.

To date, few studies have explored the prediction of BO after adenovirus pneumonia in pediatric patients via CT-based radiomic features. In our study, we applied a robust methodology, including the MI, mRMR, and LASSO methods, to extract and select quantitative radiomic features from the ROIs in the CT images of patients with adenovirus pneumonia. This approach allowed us to construct a radiomic prediction model for BO after adenovirus pneumonia. Notably, our study identified 10 optimal radiomic features closely linked to the occurrence of BO. These features included two first-order features, Seven texture features, and one shape feature, encompassing three features from GLSZM and one from GLDM, along with three from the GLCM. Notably, nearly 96% of these selected features exhibited high levels of stability and repeatability, as indicated by their interclass correlation coefficient exceeding 0.85 during redelineation. This stability suggested that the radiomic features used in our study were reliable and could be used with confidence for predictive purposes.

Our study highlighted the importance of specific radiomic features. The maximum 2D diameter column, derived from shape features, effectively characterizes the overall sizes of pulmonary inflammatory lesions and was speculated to be associated with the development of BO following adenovirus pneumonia. First-order features, such as the median and root mean squared, are commonly used to describe the signal intensity distribution, which cannot capture the spatial relationship between voxel features (Mayerhoefer et al., 2020; Peng et al., 2022). To explore heterogeneous tissue distributions and their correlations, texture features or filtered conversions are necessary to highlight or emphasize certain frequency domains (Peng et al., 2022). Texture features can reflect signal intensity heterogeneity within a lesion. Different radiomic feature parameters, such as the GLCM, GLSZM, and GLDM, capture image information in various ways, considering aspects such as change amplitude, adjacent interval, direction, homogeneous region characteristics, and gray-level correlation (Mayerhoefer et al., 2020). This multifaceted approach to radiomics enables the identification of findings that might be challenging to detect through conventional imaging examinations. By employing quantitative radiomic evaluation, our study offers a more objective and realistic approach for predicting BO after adenovirus pneumonia.

Previous studies (Homayounieh et al., 2020; Purkayastha et al., 2021) have demonstrated that the combined model has superior performance compared with both CT radiomics-based and clinical models in predicting disease progression and prognosis of COVID-19 patients. In our study, the combined models constructed by integrating the radiomic and clinical models also demonstrated a significant improvement in predictive performance, as evidenced by the Delong test and the NRI and IDI analyses. These findings highlight the potential of the combined model as a valuable imaging biomarker for predicting BO after adenovirus pneumonia, suggesting the potential of early risk assessment and informed treatment decisions.

In this research, we conducted a comparison of the performance of three distinct machine learning models in predicting BO following adenovirus pneumonia. The LR machine learning model performed exceptionally well with the AUC (95% CI) was 0.890 [0.792–0.988], the sensitivity was 0.810, the specificity was 0.897, the accuracy was 0.860, the PPV was 0.850, the NPV was 0.867, and the F1 score was 0.829, outperforming the other two models. The LR model is a classification model used for classification problems in supervised learning. LR uses a linear model to fit the decision boundary and then obtains the probability distribution. It does not need to assume the data distribution when fitting the decision boundary, thus avoiding the problem caused by the inaccurate assumption of the data distribution. The LR model not only predicts categories but also provides the probability distribution for these predictions which can be useful for tasks that require probability for decision making purposes. The LR model is computationally simple and interpretable, and shows good performance when handing high-dimensional data. Therefore, combining the advantages of the LR model with the results of our study, we believe that the LR model has important practical value in predicting BO after adenovirus pneumonia in children.

Nomograms have seen widespread use for assessing disease risk and prognosis (Wu et al., 2020; Zhang et al., 2021). To facilitate clinical decision-making, we developed a nomogram by using two independent risk factors, the number of pneumonia lobes and the length of hospital stay, and Radscore based on LR machine learning model, which provides a visual and personalized statistical prediction tool and enhances the applicability of our study.

Our study has some limitations. Being a retrospective study, there may be inherent bias in case selection. The strict inclusion criteria and potential loss to follow-up led to a relatively small sample size of 165 patients. Future research could conduct larger, multicenter studies with external validation to improve the generalization ability of the model and promote the wide application of the model in clinical practice. Additionally, the manual delineation of the ROI on nonenhanced images introduces the potential for variations in accuracy on the basis of radiologist experience. Finally, the utilization of CT scans from different manufacturers during this study may have impacted the predictive performance of the radiomic model.

Conclusions

Our study emphasizes the potential of combining CT-based radiomic features with clinical parameters to predict BO following adenovirus pneumonia in children. Our analysis of machine learning models confirms the significance of the LR model in predicting BO after adenovirus pneumonia. In future research, we aim to increase sample size and optimize the model to fully exploit its predictive capabilities in clinical practice, offering guidance for treatment and prevention.

Supplemental Information

Supplemental Information 1 Table of selected optimal radiomics features.

Supplemental Information 2 STROBE Statement.

Supplemental Information 3 Raw data.

Supplemental Information 4 In-depth introduction to this research platform as well as the data processing procedure.

(1) The original image prior to ROI labeling; (2) Images post-ROI labeling; and (3) Label instructions. As the uAI research portal (uRP) is a business intelligence research platform, the computer code is currently unavailable due to commercial interests. We have added a literature to the manuscript (the 23rd article) and attached the full text of the literature, which provides an in-depth introduction to this research platform as well as the data processing procedure.

Additional Information and Declarations

Competing Interests

Fang Wang, Man Li and Yang Li are employed by Shanghai United lmaging Intelligence Co., Ltd.

Author Contributions

Li Zhang conceived and designed the experiments, performed the experiments, prepared figures and/or tables, authored or reviewed drafts of the article, and approved the final draft.

Ling He performed the experiments, authored or reviewed drafts of the article, and approved the final draft.

Guangli Zhang performed the experiments, authored or reviewed drafts of the article, and approved the final draft.

Xiaoyin Tian analyzed the data, prepared figures and/or tables, and approved the final draft.

Haoru Wang analyzed the data, prepared figures and/or tables, and approved the final draft.

Fang Wang analyzed the data, prepared figures and/or tables, and approved the final draft.

Xin Chen performed the experiments, prepared figures and/or tables, and approved the final draft.

Yinglan Zheng performed the experiments, authored or reviewed drafts of the article, and approved the final draft.

Man Li analyzed the data, authored or reviewed drafts of the article, and approved the final draft.

Yang Li analyzed the data, prepared figures and/or tables, authored or reviewed drafts of the article, and approved the final draft.

Zhengxiu Luo conceived and designed the experiments, prepared figures and/or tables, authored or reviewed drafts of the article, and approved the final draft.

Human Ethics

The following information was supplied relating to ethical approvals (i.e., approving body and any reference numbers):

The research received approval from the Ethics Committee of Children’s Hospital of Chongqing Medical University (No. 2023–151).

Data Availability

The following information was supplied regarding data availability:

The raw data is available in the Supplemental File.

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
