# Peer review of "Computed tomography-based radiomic features combined with clinical parameters for predicting post-infectious bronchiolitis obliterans in children with adenovirus pneumonia: a retrospective study"

_PeerJ, doi:10.7717/peerj.19145_

## Round 0.1 · original submission · Major Revisions

After careful consideration of the reviewers' feedback, major revisions are required before your manuscript can be considered for publication. Please address all concerns raised by the reviewers, with particular attention to Reviewer #3's major comments. When submitting your revised manuscript, please include a detailed point-by-point response addressing how you have incorporated each reviewer's suggestions.

Reviewer 1 ·

Basic reporting

Computed tomography-based radiomic features combined with clinical parameters for predicting post-infectious bronchiolitis obliterans in children
with adenovirus pneumonia
Li Zhang, Ling He, Guangli Zhang, Xiaoyin Tian, Haoru Wang, Fang Wang, Xin Chen, Yinglan Zheng, Man Li, Yang Li, Zhengxiu Luo

Zhang and colleagues worked on the clinical and scientific context of bronchiolitis obliterans (BO) and its association with adenovirus pneumonia in children. It provides a clear rationale for exploring radiomics as a predictive tool for bronchiolitis obliterans (BO) addressing a critical gap in pediatric pulmonology. Predicting BO is novel and highlights the potential of emerging technologies in pediatric respiratory diseases. This research demonstrates the potential of integrating advanced imaging analytics with clinical data to enhance predictive accuracy for complex pediatric conditions.

Experimental design

1. Authors can explain why children are more susceptible to post-infectious bronchiolitis obliterans (PIBO) than adults. Is it due to lungs structure ? please mention.
2. In lines 66-68 authors mentioned its use in adult lung diseases and mention how its application to pediatric lung diseases then it may specifically justify your study.
3. Why is the enrollment period (October 2013 to February 2020) mentioned? Was it because of the availability of the data from that particular period of time?

Validity of the findings

4. Please provide a more detailed explanation about how radiomics scores were calculated to maintain in reproducibility.
5. Mention the threshold values for specific terms such as “fever” or “peak temperature”.
6. Please provide the significance of ICC threshold. Why it is so important.
7. Please describe preprocessing techniques such as isotropic voxel resampling and intensity normalization.
8. Please include details on hyperparameter tuning for the machine learning algorithms (For Example: SVM and RF)
9. In study, out of 2116 patients, 165 met the study criteria, divided into BO (n=70) and non-BO (n=95) groups. Is this sample size is sufficient to make the study conclusions. Please explain

Annotated reviews are not available for download in order to protect the identity of reviewers who chose to remain anonymous.

Reviewer 2 ·

Basic reporting

This study aimed to develop a non-invasive model for predicting post-infectious bronchiolitis obliterans (BO) in children with adenovirus pneumonia by integrating computed tomography (CT)-based radiomic features and clinical parameters. The study retrospectively enrolled 165 children, 70 with BO and 95 without, and stratified them into training and testing groups. Radiomic features were extracted from segmented lesions in baseline CT images, with the optimal features incorporated into combined models with clinical variables (length of hospitalization and number of pneumonia lobes) using logistic regression, random forest, and support vector machine algorithms. The combined models demonstrated high predictive performance, achieving AUCs greater than 0.85 in both training and testing groups, outperforming models solely based on radiomic or clinical features. This suggests that integrating CT-based radiomic features with clinical parameters provides an effective non-invasive approach for predicting BO in children with adenovirus pneumonia. However, there are still many areas for improvement in this study.
1. While the introduction describes the challenges of BO diagnosis and the need for improved prediction models, it lacks a clear and concise problem statement. Explicitly stating the limitations of existing clinical models and emphasizing the need for a more accurate predictive approach would strengthen the argument for the study's significance.
2. The introduction briefly mentions radiomics, but could benefit from a more detailed explanation of its principles and potential advantages in this specific context. Explaining how radiomic features relate to BO development and how they might offer unique insights beyond traditional clinical parameters would further justify the study's rationale.
3. The introduction mentions that few studies have explored radiomics for BO prediction after adenovirus pneumonia, but doesn't clearly state the specific gap this study aims to fill. For example, it could highlight the lack of a model incorporating both radiomic and clinical features, or emphasize the need for a model with improved sensitivity compared to existing approaches.
4. The inclusion criteria for adenovirus pneumonia seem relatively broad. Providing more specific details about the diagnostic criteria used (e.g., chest X-ray findings, clinical presentation) would enhance the clarity of the study's population. The criteria for diagnosing BO after adenovirus pneumonia mention "more than 6 weeks" of symptoms, but it's unclear when these symptoms were assessed relative to the acute adenovirus pneumonia episode. Specifying the time frame would clarify the temporal relationship and potential impact on the diagnosis.
5. The study mentions using different CT scanners with varying parameters. While the scanning parameters are outlined, it's unclear whether these protocols were standardized across all patients or if any adjustments were made to account for differences in scanners. Standardization of CT protocols is crucial for ensuring comparability of images and minimizing potential bias.
6. The results section mentions that AUC values are shown in Figure 3, but doesn't provide any further details about the figure. Adding a brief description of the figure, including the types of plots used to visualize the model performance (e.g., ROC curves), would make the results section more informative and engaging.
7. While the Results mentions AUC, it would be beneficial to include other important performance metrics for the radiomic models, such as sensitivity, specificity, and precision for both the training and testing cohorts. This would provide a more comprehensive assessment of the models' accuracy and reliability.
8. The Results in Figure 5 mentions the calibration curve and DCA, but lacks specific details about these analyses. Including a brief description of the methods used for calibration curve analysis and the criteria for evaluating the DCA results would make the results more informative.
9. The discussion mentions the study's limitations related to sample size and retrospective design, but doesn't extensively discuss the potential impact on the model's generalizability. Expanding on this point, including a discussion of potential factors that might limit the model's applicability to other populations and settings, would be beneficial.
10. While the discussion mentions the LR model's performance, it doesn't thoroughly explain the rationale for selecting this model over the other models (RF and SVM). Highlighting the specific advantages of LR for this particular study, such as its interpretability or suitability for handling high-dimensional data, would strengthen the argument for its use.
11. The discussion mentions that Li Peng et al. developed a nomogram prediction model for PIBO with an AUC of 0.857. However, it doesn't directly address any potential differences or conflicts between the two models. Acknowledging these differences and discussing potential explanations, such as variations in the inclusion criteria or the specific variables analyzed, would enhance the discussion's objectivity.
12. To enhance the reader's engagement, the manuscript could benefit from a language edit. Some sentences could be simplified for better clarity, and the figures and legends could be made more detailed to provide a deeper understanding of the data.

Experimental design

Merge in the Basic reporting

Validity of the findings

Merge in the Basic reporting

Reviewer 3 ·

Basic reporting

The manuscript structure is complete, and there is also a standardized format and language. Raw data can be accessed.

Experimental design

Please see Additional comments.

Validity of the findings

Please see Additional comments.

Additional comments

The manuscript has some research value, but there are many questions that need to be clarified by the author:
1.Ignoring here that the proposed model is to mimic a physician deriving a diagnosis of BO vs. no BO, the manuscript reports standalone performance measures of developed models. It is unclear how such model would be placed in clinical practice. Is the intended use of the model to replace a physician? Is the intended use of such model to supplement a physician? The authors should specify such. In the latter case of supplementing a physician diagnosis, one would further need to understand if the developed model would actually improve diagnosis performance of a physician determining whether a patient is or is not subject to bronchiolitis obliterans (BO) after adenovirus pneumonia.
2.The study relies on a very specific separation of the dataset into a training cohort and a testing cohort. Such chosen separation is not only limited, but insufficient to demonstrate model performance. Noting that such is frequently ground for sampling bias. When using machine learning model, it is quite critical to consider a cross-validation mechanism (such as the Leave-One-Out Cross-Validation) to estimate and report reliable and unbiased model performance. The study dataset is quite small and would allow use of the Leave-One-Out Cross-Validation. Using such mechanism, feature extraction and model training is performed on a n-1 cases (with n being the total number of cases) and the remaining left out case is used as the test point (for the performance estimation). This is repeated n times with each time having a different test case left out of the feature extraction and model training set.
3.unclear if all data was used in the univariate and multivariate logistic analysis (refer to Table 2 Univariate and Multivariate logistic analysis in the cohort)
4.unclear impact of data preprocessing techniques (how were these techniques decided and selected? Were these techniques set using all data (any preprocessing parameters set on all the data)?
5.Important comparisons between pairs of models on the training cohort are missing. It is unclear how different the results of these models are. The authors should consider improving analysis of their data and results.
6.Three types of machine learning models, i.e., Logistic Regression (LR), Random Forest (RF), and Support Vector Machine (SVM) methods were used for model construction. The authors state for the combined (Clinical + Radscore) model: "In the testing cohort, the LR machine learning model performed exceptionally well …". The selection as determined by the author is biased. If the study is to determine the specific machine learning model in an unbiased manner, one would need to determine such on the training cohort's result and not that of the testing cohort. In fact, when doing so, the Random Forest (RF) model would be selected as the best model to predict BO and not the Logistic Regression (LR) model.
7.Criteria for the diagnosis of PIBO are described, including pulmonary function (PF) results. You do not mention which FP tests were performed or the age at BOPI diagnosis. The patients included for the analysis have a mean age of 20 m. How many of them had done PFT?

---

## Round 0.2 · accepted · Accept

All three reviewers have endorsed the revisions, confirming that their concerns have been adequately addressed. We appreciate your thorough responses and the improvements made to the manuscript. Congratulations on this achievement!

Reviewer 1 ·

Basic reporting

Zhang and colleagues answered or clarified the details mentioned in the initial review. There are no additional comments or suggestions to provide.

Experimental design

Zhang and colleagues answered or clarified the details mentioned in the initial review. There are no additional comments or suggestions to provide.

Validity of the findings

Zhang and colleagues answered or clarified the details mentioned in the initial review. There are no additional comments or suggestions to provide.

Reviewer 2 ·

Basic reporting

no comment

Experimental design

no comment

Validity of the findings

no comment

Reviewer 3 ·

Basic reporting

The author largely addressed my concerns.

Experimental design

The author largely addressed my concerns.

Validity of the findings

The author largely addressed my concerns.